

# Public perception of the vegetative state/unresponsive wakefulness syndrome: a crowdsourced study

Daniel Kondziella[1,2,3], Man Cheung Cheung[4] and Anirban Dutta[4]

[1] Department of Neurology, Rigshospitalet, Copenhagen University Hospital, Copenhagen, Denmark
[2] Faculty of Health and Medical Sciences, University of Copenhagen, Copenhagen, Denmark
[3] Department of Neuroscience, Norwegian University of Science and Technology, Trondheim, Norway
[4] Department of Biomedical Engineering, University at Buffalo, The State University of New York (SUNY), Buffalo, NY, United States of America

## ABSTRACT

**Background**. The vegetative state (VS)/unresponsive wakefulness syndrome (UWS) denotes brain-injured, awake patients who are seemingly without awareness. Still, up to 15% of these patients show signs of covert consciousness when examined by functional magnetic resonance imaging (fMRI) or EEG, which is known as cognitive motor dissociation (CMD). Experts often prefer the term *unresponsive wakefulness syndrome* to avoid the negative connotations associated with *vegetative state* and to highlight the possibility for CMD. However, the perception of VS/UWS by the public has never been studied systematically.

**Methods**. Using an online crowdsourcing platform, we recruited 1,297 lay people from 32 countries. We investigated if *vegetative state* and *unresponsive wakefulness syndrome* might have a different influence on attitudes towards VS/UWS and the concept of CMD.

**Results**. Participants randomized to be inquired about the *vegetative state* believed that CMD was less plausible (mean estimated frequency in unresponsive patients 38.07% ± SD 25.15) than participants randomized to *unresponsive wakefulness syndrome* (42.29% ± SD 26.63; Cohen's d 0.164; $p = 0.016$). Attitudes towards treatment withdrawal were similar. Most participants preferred *unresponsive wakefulness syndrome* (60.05%), although a sizeable minority favored *vegetative state* (24.21%; difference 35.84%, 95% CI 29.36 to 41.87; $p < 0.0001$). Searches on PubMed and Google Trends revealed that *unresponsive wakefulness syndrome* is increasingly used by academics but not lay people.

**Discussion**. Simply replacing *vegetative state* with *unresponsive wakefulness syndrome* may not be fully appropriate given that one of four prefer the first term. We suggest that physicians take advantage of the controversy around the terminology to explain relatives the concept of CMD and its ethical implications.

Corresponding author
Daniel Kondziella,
daniel_kondziella@yahoo.com

## INTRODUCTION

The term *vegetative state* (VS) was coined in the 1970'ies to describe a condition of wakefulness without awareness following brain injury (*Jennett & Plum, 1972*). Patients in VS may open their eyes but exhibit only reflex behaviors during clinical examination and were therefore considered unaware of themselves and their surroundings. The word *vegetative* (referring to the preserved autonomous nervous system) is etymologically related to *vegetable*, which may evoke negative associations. Thus, in 2009, the European Task Force on Disorders of Consciousness introduced the term *unresponsive wakefulness syndrome* (UWS) (*Laureys et al., 2010*).[1] It was felt that this term lacked the pejorative connotations of *vegetative state* and enabled the medical community to refer to the level of consciousness "in a human way" (*Von Wild et al., 2012*).

Aside from the semantic peculiarities of VS/UWS, in the 2000s neuroscientists and physicians began to recognize a disturbing dilemma: Brain-injured patients who appear entirely unresponsive at the bedside may show signs of covert consciousness when examined by functional MRI (fMRI) or EEG (*Schnakers et al., 2008*; *Bekinschtein et al., 2009*; *Monti et al., 2010*; *Bruno et al., 2011*; *Boly et al., 2011*; *Faugeras et al., 2011*; *Cruse et al., 2011*; *Casali et al., 2013*; *Formisano, D'Ippolito & Catani, 2013*; *Sitt et al., 2014*; *Stender et al., 2014*; *Di Perri et al., 2016*; *Edlow et al., 2017*; *Chennu et al., 2017*; *Bodien, Chatelle & Edlow, 2017*; *Curley et al., 2018*; *Engemann et al., 2018*). About 15% of behaviorally unresponsive patients can participate in mental tasks by modifying their brain activity during EEG- or fMRI-based paradigms, suggesting that they are conscious and misdiagnosed (*Kondziella et al., 2016*). In 2015, this state of covert consciousness was termed *cognitive motor dissociation* (CMD) (*Schiff, 2015*).

Many experts prefer the term *unresponsive wakefulness syndrome* to avoid the negative connotations associated with *vegetative state* and to highlight the possibility for CMD (*Gosseries et al., 2011*; *Machado et al., 2012*; *Von Wild et al., 2012*; *Wannez et al., 2017a*; *Vanhaudenhuyse et al., 2018*), while others have been skeptical because the adjective unresponsive is open for ambiguous interpretations (*Naccache, 2018*; *Giacino et al., 2018*). However, the perception of VS/UWS by the public has never been examined systematically.

Here, we recruited a large global sample of lay people to evaluate public perception of the VS and UWS. Specifically, we tested the hypothesis that *vegetative state* and *unresponsive wakefulness syndrome* might have a different influence on lay people's attitudes towards treatment withdrawal and understanding of the CMD concept. In addition, we searched Google Trends and PubMed between 2004–2018 to compare trends in the application of the two terms by lay people and academics, respectively.

## METHODS

### Hypotheses and research questions

We aimed to test the following research questions:

- Is *vegetative state* associated with a greater likelihood of people agreeing with treatment withdrawal as compared to *unresponsive wakefulness syndrome*?

[1] In the text, we use "*vegetative state*" and "*unresponsive wakefulness syndrome*" (in italics) when referring to semantics, and "VS/UWS" when referring to the medical condition.

**Table 1** **Case history and written instructions to participants of the first sub-study (part I), where we tested if the terms *vegetative state* and *unresponsive wakefulness syndrome* might influence attitudes towards treatment withdrawal or the perception of how frequent cognitive motor dissociation (CMD) is.** Participants were randomized into two groups: The first group was given the text as outlined here. The second group received the same text and the same instructions except that *vegetative state* was replaced by *unresponsive wakefulness syndrome*.

Patient M. is a 25-year old woman with a brain injury after being hit by a car 2 years ago. She is now in a **vegetative state**. She resides in a nursing home. During daytime, the nurses place her in a chair. Her eyes are open, but she does neither look at the nursing staff nor at visitors, including family. She does not say anything, nor does she make any comprehensible sounds. When being touched or talked to, she does not react in any discernable way. She has a feeding tube inserted into her stomach, and she is incontinent for urine and stool.

Please indicate how strongly you agree or disagree with the following statements about the **vegetative state**:

- *Following detailed discussions with the relatives of a patient in the **vegetative state**, it is morally acceptable to end the patient's life by withdrawing treatment if there is no medical hope for recovery.*[a]
- *I would prefer treatment withdrawn if I were in a **vegetative state** without hope for recovery.*[a]

Research using modern neuroimaging techniques shows that some patients who fulfill the clinical criteria for the **vegetative state** nevertheless are conscious to such a degree that they can think or imagine things, e.g., thinking of playing tennis or imagining walking around in a house. Thus, although these patients look completely unaware when examined at the bedside, and are unable to communicate in any way, modern technologies reveal that some of them still show evidence of a rich mental life.

- *For how large a percentage of patients with the **vegetative state** would you believe this is true? Please state your best estimate (from 0% to 100%).*[b]

**Notes.**
[a] A 7-point Likert scale was employed, ranging from "strongly agree" to "strongly disagree", to allow participants to express uncertainty.
[b] Participants were asked to state their estimate using a visual analog scale from 0–100.

- Would people more easily agree with treatment withdrawal in a hypothetical case involving themselves than they would in a case involving someone else?
- Are religious people more likely to disagree with treatment withdrawal?
- Is *unresponsive wakefulness syndrome* associated with a greater likelihood of people believing that a patient might be in a state of CMD as compared to *vegetative state*?
- If asked directly, do people prefer *unresponsive wakefulness syndrome* over *vegetative state*?

## Study design

An online platform, Prolific Academic (https://prolific.ac/), was used to recruit lay people for two sub-studies (part I and II). Prolific Academic is a crowdsourcing platform dedicated to recruit online human subjects for research that has been shown to compare favorably with Amazon's Mechanical Turk in terms of honesty and diversity of participants, as well as overall data quality (*Woods et al., 2015*; *Peer et al., 2017*). Participants were recruited without any filters except for age ≥18 years and English language.

For part I, we asked participants to complete a questionnaire comprising demographic information on age, educational background, degree of religiosity (using a 7-point Likert scale, ranging from "extremely important" to "not at all important") and place of residence. Participants were then randomized into two groups: Both groups were given a fictive case history of a VS/UWS patient followed by questions about attitudes towards withdrawal of treatment. In addition, participants were asked to estimate the likelihood (from 0–100%) of CMD in a given VS/UWS patient, as well as if they knew someone close to them being in VS/UWS. The case history and questions were similar for both groups except that we used the term *vegetative state* for the first group and the term *unresponsive wakefulness syndrome* for the second group. See Table 1 for details.

For part II, we recruited another sample of participants (participants from part I were excluded). Following the same fictive case history as in part I (yet without mentioning of *vegetative state* or *unresponsive wakefulness syndrome* in the text), participants were asked which of the two terms they would prefer doctors, friends and family to use, if they (the participants) or a loved one were in the same state as the fictive patient. Hence, here we assessed preferences directly, without randomization. In addition, we asked if participants had heard about VS, UWS or both before this survey.

## Statistics

For part I, which included randomization of participants into two groups, we calculated the total number of participants required to be 858, using a small effect size (0.2), a one-tailed *t*-test (since we expected *vegetative state* to have a unidirectional, i.e., negative, effect), a high power (0.9) and a significance level ($\alpha$) of 0.05. For part II, we estimated the number of participants required to be 384, using a very high population size (300.000.000), a confidence level of 95% and a margin of error of 5%.

Data were analyzed according to standard procedures using IBM SPSS Statistics 22 (Armonk, NY, US). Effect sizes were calculated according to *Lakens (2013)*. To prevent HARKing (Hypothesizing After the Results are Known) (*Fraser et al., 2018*), we pre-registered the study, including all hypotheses, with the Open Science Framework (https://osf.io/kr3q9/).

## Ethics

Participants gave consent for publication of their (anonymous) data. Participation was anonymous, voluntary and restricted to those $\geq$18 years. Participants received a monetary reward upon completion of the survey, following the platform's *ethical rewards* principle ($\geq$ \$6.50/h). The Ethics Committee of the Capital Region of Denmark waives approval for online surveys (Section 14 (1) of the Committee Act. 2; http://www.nvk.dk/english).

## Searches using PubMed and Google Trends

We searched PubMed for papers on *vegetative state* and *unresponsive wakefulness syndrome* (restricted to human subjects) for the period January 2004–November 2018 according to standard bibliographic methods (key words ''vegetative state'' and ''unresponsive wakefulness syndrome'', filter ''humans''). Further, we searched Google Trends for the identical period using the same search terms (''vegetative state'', ''unresponsive wakefulness syndrome'') for global internet searches, expressed as Google Relative Search Volumes (ranging from 0–100).

## RESULTS

We recruited 1,297 participants from 32 countries (mean age 32.06 years $\pm$ SD 10.12; $n = 884$ for part I, $n = 413$ for part II), most of which were well-educated with a secular background and residing in Europe and North America. Table 2 and Fig. 1A provide epidemiological information. Raw data are provided in the *online supplemental files.*

More participants had heard about *vegetative state* (91.83%) than *unresponsive wakefulness syndrome* (18.56%; difference 73.27%, 95% CI 62.52 to 80.84, Chi-squared
**Table 2 Demographic information on 1,297 lay people from 32 countries, including Europe (United Kingdom, 560 participants; Poland, 100; Portugal, 83; Spain, 77; Italy, 74; Germany, 45; Netherlands, 29; Greece, 17; Hungary, 17; Sweden, 14; Estonia, 10; France, Finland, Ireland, Belgium, Slovenia, Switzerland, Austria, Latvia, Czech Republic, Norway, Denmark, and Iceland, all <10), the Americas (US, 57; Canada, 38, Mexico, 29, Chile 9), the Middle East (Israel, 7; Turkey, 5), Asia (Japan, 3; South-Korea, 2), and Australia (23).** The place of residency of 37 participants was unknown.

| Place of residence (n = 1,297)[a] | Gender (n = 1297) | Age (n = 884) | Education (n = 884) | Employment (n = 1,297) | Religiosity (n = 884) |
|---|---|---|---|---|---|
| Europe (n = 1088) | Male | Mean | University or higher | Full-time | Extremely important |
| UK: 560, 43.2% | 654, 50.4% | 32.06 ± SD 10.12 years | 485, 54.9% | 524, 40.4% | 36, 4.1% |
| Other: 528, 40.7% | | | | | Very important |
| Americas (n = 132) | Female | Median | High school and/or college | Half-time/other | 86, 9.7% |
| US: 57, 4.39% | 643, 49.6% | 30 years (range 18–86) | 380, 43.0% | 617, 47.6% | Somewhat important |
| Other: 75, 5.78% | | | | | 117, 13.2% |
| Australia: 23, 1.77% | | | Less than high school | Unemployed | Not so important |
| Middle East and Asia | | | 19, 2.1% | 156, 12.0% | 186, 21.0% |
| Middle East: 12, 0.93% | | | | | Not at all important |
| Asia: 5, 0.39% | | | | | 459, 51.9% |
| Unknown: 37, 2.85% | | | | | |

**Notes.**

[a] n, number of participants with available data.

205.11; $p < 0.0001$). Only few participants had not heard of either term (4.46%). A minority of participants (4.52%, from total $n = 413$) had personal experience of someone close to them being in VS/UWS.

Results are listed according to the pre-specified five research questions as outlined in Methods.

- *Attitudes towards treatment withdrawal: vegetative state vs. unresponsive wakefulness syndrome Vegetative state*-was not associated with greater likelihood of participants agreeing with treatment withdrawal (48.79%) as compared to *unresponsive wakefulness syndrome* (46.47%; total $n = 880$; relative risk 1.0520, 95% CI 0.9143 to 1.2104; $p = 0.48$). The average score of participants on the 7-point Likert scale (1 = "strongly agree with treatment withdrawal", 7 = "strongly disagree") was 2.87 ± 1.59 for vegetative state and 2.94 ± 1.63 for *unresponsive wakefulness syndrome*. Equivalence testing, using a small effect size (0.2), was significant, $t(817.44) = 2.301$, $p = 0.0108$, whereas the null hypothesis test was non-significant, $t(817.44) = -0.639$, $p = 0.523$. Based on the equivalence test and the null-hypothesis test combined, we can conclude that any difference in the effect size of *vegetative state* and *unresponsive wakefulness syndrome* on attitudes towards treatment withdrawal must be smaller than 0.2.
- *Attitudes towards treatment withdrawal: oneself vs. someone else* Participants did more often agree with treatment withdrawal in a hypothetical case involving themselves being in VS/UWS (32.27%) than in a case with someone else being in VS/UWS (19.20%; total $n = 880$; difference 18.07%, 95% CI 13.91 to 22.14, Chi-squared 70.863; $p < 0.0001$) (Fig. 1B).
- *Attitudes towards treatment withdrawal: high vs. low degree of religiosity* Compared to participants with the lowest degree of religiosity (51.92%), participants with the highest

<br>

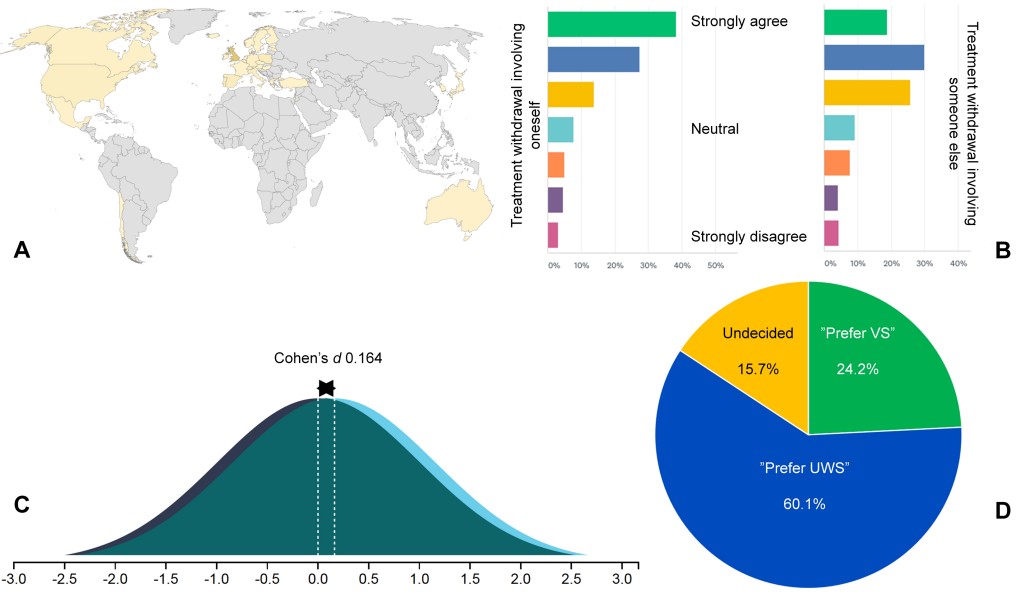

**Figure 1** **Overview of study methods and main results.** Using an online crowdsourcing platform, we recruited 1,297 participants from 32 countries on five continents, the majority from Europe and North America (A). Most participants strongly agreed with treatment withdrawal in a hypothetical case involving themselves being in VS/UWS, but the participants had more concerns if they had to decide on treatment withdrawal for someone else (B). Participants who read a case history of a fictive patient in the *vegetative state* were less likely to believe in the possibility of cognitive motor dissociation than those who read about the same patient with the *unresponsive wakefulness syndrome*, a small but statistically significant effect (Cohen's d 0.164; $p = 0.016$) (C). When asked directly, most participants preferred the *unresponsive wakefulness syndrome* (blue), although nearly one in four favored *vegetative state* (green) (D).

degree of religiosity (4.07%) were more likely to be undecided or against treatment withdrawal (20 of 36 religious participants vs. 113 of 459 secular participants; relative risk 1.5963, 95% CI [1.1257–2.2636]; $p = 0.0087$).

- *Plausibility of the concept of cognitive motor dissociation* Participants exposed to the term *vegetative state* believed that the average % of unresponsive patients with CMD was lower (median % 35, mean % 38.07 ± SD 25.15) than participants exposed to the term *unresponsive wakefulness syndrome* (median % 40, mean % 42.29 ± SD 26.63; total $n = 880$), a small but significant difference (Cohen's d 0.164; $p = 0.016$) (Fig. 1C).

- Preferences for *unresponsive wakefulness syndrome versus vegetative state* Most participants preferred *unresponsive wakefulness syndrome* (60.05%), whereas a sizeable minority of respondents favored *vegetative state* (24.21%; difference 35.84%, 95% CI 29.36 to 41.87, Chi-squared 108.66; $p < 0.0001$). Sixty-five respondents (15.74%, total $n = 413$) were undecided (Fig. 1D).

Selected comments from participants on their preference for one term over the other can be found in Tables 3 and 4.

A search on PubMed showed a clear trend for the increasing use of *unresponsive wakefulness syndrome* in medical papers as compared to *vegetative state*. In 2012, for instance, there were 13 papers on *unresponsive wakefulness syndrome* compared to 103

**Table 3** Selected comments from participants on their preference for the term *unresponsive wakefulness syndrome.* Comments are edited for clarity and spelling.

- [VS] sounds dehumanizing, as if I was demeaning a loved one.
- [VS] sounds horrible, like you're alive in only the same way as a plant and not a human.
- [UWS] is less insulting and more accurate.
- [VS] makes me think of vegetables, which is deeply offensive. [UWS] is a lot more respectful and you never know if the patient can hear you talking.
- [VS] sounds really depressing, and I would feel very upset if my doctor would use it.
- [UWS] sounds kinder and respectful.
- [VS] is just a step away from the derogatory slang "vegetable".
- [UWS] is more gentile.
- [UWS] is a bit less dehumanizing. Also, it feels more medical and as if it could be reversed.
- [UWS] sounds more "professional", more serious. Words like "vegetative state" are insulting because they make it sound as if the patient was an object.
- [VS] suggests that these patients are less than human. I wouldn't want someone I love being described as such.
- [VS] takes away all dignity and humanity.
- With our current level of medical knowledge, we should never refer to someone who is awake and living as "vegetative".
- [UWS] emphasizes that the patient is "awake" even if not currently responsive.
- [UWS] sounds less harsh and explains the condition better.
- [UWS] gives more hope for the patient to recover.
- [VS] sounds very negative to me, it's like being compared to a plant.
- [UWS] sounds nicer and I believe everyone deserves respect.
- [VS] sounds harsh.
- [UWS] is more politically correct and sounds more professional. The term vegetable has been historically used as a derogatory term.
- [VS] is extremely direct, pejorative, while its euphemism [UWS] is much more polite and tactful.
- [UWS] feels more human, more emotional, like the person she was.
- [VS] seems pedantic, dehumanizing when they're still alive.
- I believe that the eyes being open is a sign of "wakefulness".
- [VS] reminds me of when it was socially acceptable to call people vegetables.
- If I were able to still understand what I was hearing, I would prefer not to be compared to a vegetable since I wouldn't have much else going for me at that time.
- [VS] sounds awful. It's really comparing you to an inanimate object.
- [VS] is insulting, referring to a vegetable and not acknowledging who the person was before.
- My mum was in this state for a period, she would have been extremely upset to hear people say [VS]. She is a human and should be referred to with respect, not like a vegetable.
- You get shocked by [VS]. I believe that calling it [UWS] takes away the severity of the situation.
- [VS] is more used by the media, so it has more negative feelings attached to it. [UWS] feels almost as if there was a cure.
- [VS] just sounds weird and offensive and is not very professional.
- [VS] sounds more depressing, as if the person was brain-dead.
- After watching some stories on Youtube about this topic I think [UWS] sounds more fitting.
- The term vegetable has been a mocking term in childhood/youth. It also sounds harsh.
- [UWS] is a hopeful term, as if the person is still there.
- "Vegetative" is not a decent way to describe anyone. We are all humans, not plants.
- [UWS] seems more alive to me. [VS] sounds like death.

**Table 3** (*continued*)

- [VS] implies that a person is no longer human.
- [UWS] gives impression of hope—like any other medical condition.
- [UWS] is a hopeful word that sounds as if there could be some conscious activity going on. It gives hope that this is not a permanent state.
- I hate the idea of calling someone a vegetable. It sounds so old-fashioned and ignorant.
- [UWS] has more emphasis on the fact that the person is awake and only can't respond.
- [VS] sounds terrible, especially if patients can hear what is going on. It could be upsetting and frustrating to them.
- [UWS] sounds less cruel and is a more scientific term.
- Unlike [UWS], [VS] feels like there is no separation between the person and the condition and that the person is the condition.
- [VS] sounds like saying someone is a potato.
- [UWS] seems a more polite and caring way of describing the patient's condition.
- Comparing a human being with a vegetable is just unacceptable.
- [VS] feels hurtful when it concerns a loved one.
- I wouldn't like to think of a loved one as being a vegetable. Vegetables turn to mush and degrade.
- [VS] is rude and pejorative. People are not vegetables.
- As her eyes are open, [UWS] seems to fit better.
- [VS] has a history of stigmatization. It sounds unethical, offensive and mean.
- [UWS] is more explanatory in nature without relying on assumptions.
- I believe the term [UWS] gives the patient more dignity.
- Too much badness has been said about [VS], i.e., being a cabbage, hateful comments.
- [UWS] sounds like a medical term that I have got rather than what I am.
- I would like doctors to use [UWS] because it does not sound as definite as [VS].
- [VS] sounds like the patient is a 'thing' and has been given up on.
- [VS] undermines the individual's fundamental human dignity.
- [UWS] sounds a lot more professional.

papers on *vegetative state*, whereas for January–November 2018, the ratio was 62/101. In contrast, Google Trends revealed no such tendencies, but internet searches peaked around highly publicized cases in the media (Fig. 2).

## DISCUSSION

Here, we have shown that the terms *vegetative state* and *unresponsive wakefulness syndrome* have measurable effects on the way people perceive VS/UWS. Although attitudes towards treatment withdrawal appeared unchanged, people exposed to the term *vegetative state* were less likely to believe in CMD than people inquired about the *unresponsive wakefulness syndrome*. The effect was small (Cohen's $d$ 0.164), but this effect size is comparable to similar framing effects in psychological research (*Lakens, 2013*; *Open Science Collaboration, 2015*; *Koon, Hawkins & Mayhew, 2016*).

Does this effect matter for clinical routine? We believe, it does. Up to 40% of unresponsive patients are incorrectly classified as being in VS/UWS because standardized rating scales such as the Coma Recovery Scale-Revised are not widely used (*Schnakers et al., 2009*; *Di et al., 2014*; *Thonnard et al., 2014*; *Chatelle et al., 2016*; *Wannez et al., 2017a*; *Wannez et al., 2017b*; *Wannez et al., 2017c*; *Vanhaudenhuyse et al., 2018*). Moreover, an estimated 15% of patients with a clinical diagnosis of VS/UWS can follow commands by performing mental

**Table 4  Selected comments from participants on their preference for the term *vegetative state*.**  Comments are edited for clarity and spelling.

- [UWS] seems like an unnecessary euphemism.
- I think these new longer terms [like UWS] are silly.
- The name for her state [of consciousness] doesn't matter, [it] doesn't help her to get better.
- I understand [VS] instantly, but I would need an explanation for [UWS].
- I believe [VS] sounds correct from what I have seen on ER.[a]
- [VS] is well-established and generally understood; [UWS] is euphemistic.
- It is natural to call [VS] like that.
- Although it doesn't sound as nice, I think that [VS] is better understood.
- [VS] sounds a bit harsh but people in that state do seem like vegetables. It is no life anymore.
- I grew up on that term and it is shorter to say.
- [VS] seems a more appropriate term given the situation.
- [VS] is a direct message instead of euphemism. I hate ''political correctness'' and similar artificial constructs which obfuscate the meaning of the original message.
- [UWS] makes it sound like there's a potential cure—false hope.
- [UWS] would make me feel more upset. I would not like the idea that she [might be] awake.
- There's no point sugar-coating something that is inherently depressing and morbid.
- I prefer to face the facts and call things what they are.
- Although [VS] may seem harsh, it is short, precise and easily understood.
- [VS] more clearly describes the condition.
- I see [UWS] as an attempt to disguise the reality.
- [UWS] sounds overdramatic and made-up.
- The way doctors talk about the disease will not change a thing.
- ''State'' makes it sound like a short-term problem, whereas ''syndrome'' feels permanent.
- ''Vegetative'' implies vegetable, but in the medical field nobody is trying to offend—it's purely a medical term.
- [UWS] is wordy and deliberately sensitive. I feel that if this was my loved one, I wouldn't be offended by [VS].
- [VS] is less of a mouth full. [UWS] is a complicated way of saying something rather simple.
- [VS] is more commonly known, I'm familiar to the term.
- I would know immediately what it was.
- I have always used this term.
- I think the term [VS] is better known. I would feel more comfortable with that term.
- ''Syndrome'' makes any medical condition sound really bad.
- I feel like it's a little too much PC culture. I'd be okay with the term [VS] being used about my own loved ones.[b]
- [VS] seems more friendly and understandable for the average person.
- [VS] helps to avoid long explanations.
- [UWS] might be confusing and give me false hope.
- [UWS] sound like trying to sugar-coat the truth.
- [UWS] is pretentious and quite inaccurate.
- Medical conditions should be explained by using short and simple terms.

**Notes.**
[a]ER, famous American medical drama television series.
[b]PC, political correctness.

imagery tasks when examined with fMRI and/or EEG paradigms, strongly suggesting that they are indeed conscious and in a state of CMD (*Schiff, 2015*; *Kondziella et al., 2016*). This has major ethical and practical implications, including prognosis, treatment, resource

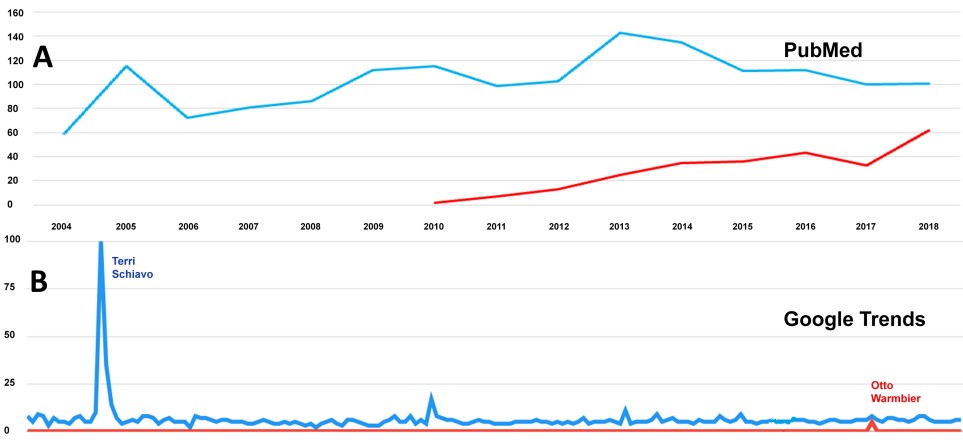

**Figure 2  Searches on PubMed and Google Trends.** While *unresponsive wakefulness syndrome* is increasingly being used by academics as shown by a PubMed search (A, papers per year; blue, papers on *vegetative state*; red, papers on *unresponsive wakefulness syndrome*), this is not the case for lay people (B, Google Relative Search Volumes, ranging from 0–100; blue, Google searches for *vegetative state*; red, Google searches for *unresponsive wakefulness syndrome*). Instead, Google searches peak around highly publicized patient cases in the media: Terri Schiavo, a patient in VS, in March 2005 (left peak, blue) and Otto Warmbier, a patient in UWS, in June 2017 (red peak, right).

allocation and end-of-life decisions (*Laureys et al., 2005*; *Turgeon et al., 2011*; *Demertzi et al., 2014*; *Di Perri et al., 2016*; *Kondziella, 2017*; *Kondziella, 2018*; *Harvey et al., 2018*; *Faugeras et al., 2018*). Careful communication about patients' levels of consciousness and the potential for CMD seems crucial in this regard (*Laureys et al., 2010*; *Gosseries et al., 2011*; *Von Wild et al., 2012*).

The present data, including the comments in Table 3, suggest that the term wakefulness in UWS seems to increase the odds of associating the VS/UWS entity with awareness. Indeed, wakefulness and awareness are frequently misinterpreted (*Naccache, 2018*), even among caregivers (*Hermann et al., 2018*). It is interesting in this regard that study participants overestimated the prevalence of covert consciousness in VS/UWS by a large margin (40%). As stated earlier, data from the scientific literature suggest that the correct rate is rather around 15% (*Kondziella et al., 2016*). Of note, our study in which lay people were explicitly primed about CMD yielded very different results compared to a previous study on the public perception of VS/UWS in which the participants were not introduced to the concept of CMD (*Gray, Knickman & Wegner, 2011*). Gray and co-workers recruited 201 lay people (51% females, mean age 23 years) from train stations and public parks in Massachusetts and New York State, USA, and asked them to read a clinical vignette involving a road traffic victim whose "entire brain was destroyed, except for the one part that keeps him breathing. So, while his body is still technically alive, he will never wake up again". Participants considered this fictive protagonist to be "more dead than dead" and with "less mental capacity than [a] dead [person]", which the authors suggested was due to "afterlife beliefs, and the tendency to focus on the bodies of [persistent] VS patients at the expense of their minds" (*Gray, Knickman & Wegner, 2011*). Again, physicians and other caregivers should

carefully discuss these discrepancies with patient relatives to achieve a realistic view of VS/UWS, while at the same time acknowledging that figures about the prevalence of CMD are likely to change in the future with more accurate data.

The many emotional comments from participants about *vegetative state* and *unresponsive wakefulness syndrome* also suggest that terminology matters. Both terms evoke strong opinions. Among the adjectives associated with the *vegetative state* were "dehumanizing", "offensive", "ignorant" and "pejorative", whereas many people related *unresponsive wakefulness syndrome* to kindness, dignity and professionalism. Of note, however, one quarter of participants favored *vegetative state* because they perceived this term as less euphemistic and easier to understand. Once more, empathic communication by the attending physicians and caregivers seems essential to navigate these controversies.

Apart from the semantic effects, our results suggest that the public attitude towards treatment withdrawal in brain-injured patients is influenced by religiosity, i.e., people with a high degree of religiosity are less likely to endorse treatment withdrawal than secular individuals (*Gipson, Kahane & Savulescu, 2014*). This corroborates previous work and is probably because values such as the sanctity of life are more important in religious groups as compared to secular groups that tend to give greater weight to patient autonomy (*Gipson, Kahane & Savulescu, 2014*). Concerns related to autonomy may also explain why most participants were even more willing to endorse withdrawal of treatment in a hypothetical case involving themselves than someone else, suggesting that the former decision is easier than the latter.

An online study such as this one has limitations that should be acknowledged (*Woods et al., 2015*; *Peer et al., 2017*). First, complex clinical and ethical notions are impossible to fully implement in a survey form. Second, it should be kept in mind that for obvious reasons only a small minority of participants (<5%) stated they personally knew someone being in VS/UWS. Although their responses did not differ from that of participants in general (see *online supplemental files*), it is obvious that people might change attitudes when being confronted with a real case of VS/UWS. Third, our sample included a high degree of secular participants; sampling from countries with strong religious cultures would likely have resulted in more people disagreeing with treatment withdrawal. Finally, for the sake of simplicity we inquired participants about their attitudes towards treatment withdrawal which is a somewhat more ambiguous term than withdrawal of life-sustaining therapies. It might be that including practical details such as stopping artificial feeding and hydration would have yielded lower approval rates. On the positive side, this is the first systematic and (and least in part I) randomized controlled study on public perception of VS/UWS and the semantic effects of *vegetative state* and *unresponsive wakefulness syndrome*. Further, we recruited a much larger sample size than what typically can be achieved with lab-based behavioral testing or mail-based questionnaires. Lastly, although we were unable to recruit participants from Africa (and although Asian participants were underrepresented), this was a truly intercontinental sample with respondents from more than 30 countries on 5 continents, which strengthens the validity and generalizability of our results.

We also retrieved publicly available data from PubMed and Google Trends to trace the application of the terms *vegetative state* and *unresponsive wakefulness syndrome* by

lay people and academics over the last 15 years. We found that *unresponsive wakefulness syndrome* is increasingly being used by academics but not by lay people, suggesting that the term is still not widely known in the public. (This is in line with our survey in which 4.5 times as many people had heard about *vegetative state*.) Instead, Google searches peaked around highly publicized patient cases in the media, most notably Terri Schiavo and Otto Warmbier. Diagnosed with a persistent VS following cardiac arrest, Terri Schiavo was the subject of a right-to-die legal case between her husband (pro treatment withdrawal) and her parents (contra), even prompting US President George W. Bush to take sides (contra). She died in March 2005, when her feeding tube was finally removed. This explains the peak in internet searches related to *vegetative state* as revealed in Fig. 2. Similarly, the case of Otto Warmbier is associated with a sharp increase (albeit much less intense) in search activities related to *unresponsive wakefulness syndrome*. Otto Warmbier was a US citizen arrested in North Korea, where he suffered a brain injury (probably due to cardiac arrest). When he returned to the US 18 months later, his physicians diagnosed "a state of unresponsive wakefulness", following which his parents requested the feeding tube to be removed. Warmbier died shortly after in June 2017. Thus, highly publicized media cases trigger public interest and may offer an opportunity to raise awareness for patients with VS/UWS and other disorders of consciousness.

## CONCLUSION

VS/UWS is a condition associated with strongly polarized opinions in the public. The term *vegetative state* has particularly negative connotations and seems counterintuitive to the concept of CMD, which is of increasing relevance given our improved abilities to identify covert consciousness in unresponsive patients using fMRI- and EEG-based paradigms. However, simply replacing *vegetative state* with *unresponsive wakefulness syndrome* may not be entirely appropriate given that one of the four still prefer the first term over the latter. Instead, we suggest that physicians and caregivers take advantage of the controversy around the terminology and use it to explain to patient relatives the concept of CMD and its inherent diagnostic uncertainties and ethical implications. This approach might reconcile any aversions the next-of-kin might have towards one or the other term and increase their understanding of the patient's condition. Ultimately, such an approach might also increase public awareness of patients with other disorders of consciousness than VS/UWS.

### Funding
The authors received no funding for this work.

### Competing Interests
The authors declare there are no competing interests.

## Author Contributions

- Daniel Kondziella conceived and designed the experiments, performed the experiments, analyzed the data, contributed reagents/materials/analysis tools, prepared figures and/or tables, authored or reviewed drafts of the paper, approved the final draft.
- Man Cheung Cheung and Anirban Dutta analyzed the data, authored or reviewed drafts of the paper, approved the final draft.

## Human Ethics

The following information was supplied relating to ethical approvals (i.e., approving body and any reference numbers):

The Ethics Committee of the Capital Region of Denmark waives approval for online surveys (Section 14 (1) of the Committee Act. 2; http://www.nvk.dk/english).

## Data Availability

The raw measurements are provided in Files S1 and S2.

## Supplemental Information

Supplemental information for this article can be found online at http://dx.doi.org/10.7717/peerj.6575#supplemental-information.

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
