# Peer review of "Public perception of the vegetative state/unresponsive wakefulness syndrome: a crowdsourced study"

_PeerJ, doi:10.7717/peerj.6575_

## Round 0.1 · original submission · Minor Revisions

Dear Authors,

Two peer reviewers have given their comments to improve the manuscript and I hope these revisions can be done soonest.

·

Basic reporting

No comment.

Experimental design

No comment.

Validity of the findings

No comment.

Additional comments

This study evaluated the different perception of “vegetative state” and “unresponsive wakefulness syndrome” among 1297 participants recruited with an online crowdsourcing platform. The paper is interesting and well-written and its results may help the physicians in their complex interaction with caregivers and relatives of patients with VS/UWS. I don’t have any major criticism but only the following suggestion to bring to the authors’ attention. The perception of the “cognitive motor dissociation” (CMD) was not explicitly evaluated in this study. However, both the abstract (discussion) and the discussion (conclusions) discuss on CMD as a direct finding of the study. My suggestion is to revise these parts of the paper, focusing only on your direct results.

Minor
Line 168: It is not clear what "total n = 404" refers to.

·

Basic reporting

Using an online crowdsourced tool, the authors explored the public perception of patients according to the terminology used to describe their state of consciousness: “vegetative state” OR “unresponsive wakefulness syndrome”.

Using a significantly large sample, they showed that:
• participants had more frequently heard about “vegetative state” (92%) than “unresponsive wakefulness syndrome” (19%).
• most participants preferred the term “unresponsive wakefulness syndrome”.
• the term “vegetative state” did not imply a higher agreement rate for withdrawal of life sustaining therapies.
• participants were slightly more likely to consider the idea that a patient can retain some conscious cognition (cognitive motor dissociation) when presented as being in an “unresponsive wakefulness syndrome” than “vegetative state” (43.3% vs 38%, respectively).


The manuscript is clear, professional English is used throughout. Background/context and literature references can be slightly improved (see General comments for the author). Structure of the article, figures and tables match the required standard. Raw data are shared and well described. The manuscript is self-contained with relevant results to hypotheses.

Experimental design

The primary research is within the Aims and Scope of the journal. The research question is well defined, relevant & meaningful. It is stated how this research fills an identified knowledge gap. Investigation was performed with a validated online tool with a high technical & ethical standards. The methods are well described with detail & information to replicate.

Validity of the findings

The data is robust, well described statistically sound, & controlled. Conclusions are well stated, linked to original research question & limited to supporting results but could be slightly improved (see General comments for the author).

Additional comments

Comment #1:
I think the statement that “most experts prefer UWS” in the abstract is over rated and should be downgraded. For instance, in the recent AAN Guidelines (Giacino Neurology 2018): “While this term has no special merit or mandate for use in clinical practice, it is included here because of its wide acceptance in Europe. ». But in Naccache Brain 2018 (from Europe): “I do not consider this proposal as a satisfactory solution, mostly because the adjective unresponsive is open to many ambiguous meanings ranging from intentional responses to reflex or automatic behavioural responses. Using the adjective ‘unresponsive’ may confuse families and relatives who often observe behavioural responses in these patients (even though only reflexive)…“.

Comment #2
I think the chosen term “treatment withdrawal” is problematic for 2 reasons:

1) It is technically the life sustaining therapies that are withdrawn (e.g. ventilator support in the acute setting and, feeding and hydration in the chronic setting). In both acute and chronic cases, treatments are delivered until death (e.g. comfort care, that can include sedation and pain management).

2) Since this “treatment withdrawal” terminology was used in the questionnaire, without further details, authors should discuss a potential misconception of what this implies (framing effect again). One could argue that a more specific framing such as for instance: “Following detailed discussions with the relatives of a patient in the vegetative state, it is morally acceptable to end the patient’s life by withdrawing treatment (including artificial feeding and hydration) if there is no medical hope for recovery.” could have led to a lower rate of acceptance for WLST.

Regarding this issue I think it is important to better clarify what kind of “treatment withdrawal” is concerned in this study. Since the clinical vignette relates a patient in a VS/UWS for 2 years, I think it only concerns the chronic setting (so mainly the artificial feeding and hydration).

Comment #3
This study mostly refers to the well-known “framing effect”, classical bias in social sciences and psychology. I think this point deserves a better discussion. Authors refer to “subliminal effect” which is a very misleading term here (especially in the field of consciousness when it mostly refers to visual masking techniques). I think this term should be removed from the entire manuscript.

Comment #4
If I properly understood, the vast majority of the participants that had heard about UWS also have heard about VS. If this is the case, the sentence “Some respondents had heard of both terms (17.33%)” is misleading and should be rephrased. This implies another bias that should be addressed: people that have heard about UWS could also most likely have heard about CMD (this should be added in the discussion).

Another related issue that could be better addressed is whether the term wakefulness in UWS could increase the odd to associate the VS/UWS entity with awareness. Indeed, “wakefulness” and “awareness” frequently are misinterpreted (even among caregivers, see for instance Hermann et all bioRxiv 2018). Similarly, the term “unresponsive” of UWS has been suggested to critics (see Comment #1and Naccache Brain 2018 “… Adopting the ‘unresponsive wakefulness’ label may even suggest to some families, relatives and caregivers of such a patient that he/she is very similar to a conscious but paralyzed patient (like for instance conceiving the patient as being in the locked-in syndrome): a conscious but unresponsive person.").

Finally, since the estimated prevalence of CMD among VS/UWS currently is 15%; authors should discuss the potential implications of such an overestimation (~40%) by lay people as well as the potential impact of the spread of the UWS terminology.

Comment #5
Please discuss potential recruitment bias and generalizability (e.g. religiosity which seems lower than average in this sample [73% “not important”]).

Minor comments:
• Abstract:
o It is not clear in the abstract that participants are lay people, please clarify
o Result: according to the Question (Table 1) I would reframe the presentation of the result as for instance: “CMD was thought to be less plausible … rather than “was less common »
• Introduction:
some references could be added for covert consciousness (e.g. Monti et al NEJM 2010, Curley et al. Brain 2018)
• Dicussion:
authors should cite and put results in perspective with previous work such as: Gray K, Anne Knickman T, Wegner DM. More dead than dead: Perceptions of persons in the persistent vegetative state. Cognition 2011;121(2):275–80.
• Line 44 & 144: provide the full html link.
• Line 103: the note “A In the text, we use “vegetative state” and “unresponsive wakefulness syndrome” (in italics) when referring to semantics, and “VS” and “UWS” when referring to the medical condition” is misleading. I would recommend to use VS/UWS for the medical condition throughout the MS.
• Line 105: “unresponsive patient” should be replaced by the unresponsive wakefulness syndrome italic notation described above (please make sure this is constant throughout the MS.
• Line 114: Add that a language filter (only English) was implicitly used. Also consider providing additional data about the online survey to help further researcher to set a similar design: how much time was required to fill in the questionnaire? What was the cost, the funding?
• Line 156: please provide the full formula used for searches on PubMed and Google Trends.
• Line 170: consider using headers for each question: e.g. Attitude towards WLST:….
• Figure 1 : color code overlap between B and D is misleading. B legend can be improved.

---

## Round 0.2 · accepted · Accept

Congratulations! Your revised manuscript has been accepted for publication in PeerJ.

Thank you and we hope that you will continue to submit future manuscripts to this journal.

# ·

Basic reporting

No comments.

Experimental design

No comments.

Validity of the findings

No comments.

Additional comments

Very interesting research, I endorse its publication.

·

Basic reporting

Using an online crowdsourced tool, the authors explored the public perception of patients according to the terminology used to describe their state of consciousness: “vegetative state” OR “unresponsive wakefulness syndrome”.

Using a significantly large sample, they showed that:
• participants had more frequently heard about “vegetative state” (92%) than “unresponsive wakefulness syndrome” (19%).
• most participants preferred the term “unresponsive wakefulness syndrome”.
• the term “vegetative state” did not imply a higher agreement rate for withdrawal of life sustaining therapies.
• participants were slightly more likely to consider the idea that a patient can retain some conscious cognition (cognitive motor dissociation) when presented as being in an “unresponsive wakefulness syndrome” than “vegetative state” (43.3% vs 38%, respectively).

The manuscript is clear, professional English is used throughout. Background/context and literature references can be slightly improved (see General comments for the author). Structure of the article, figures and tables match the required standard. Raw data are shared and well described. The manuscript is self-contained with relevant results to hypotheses.

The authors have correctly addressed all my comments. I think their manuscript has been significantly improved and is now suitable for publication.

This is a very elegant and original study that will definitely add to the field.

Experimental design

The primary research is within the Aims and Scope of the journal. The research question is well defined, relevant & meaningful. It is stated how this research fills an identified knowledge gap. Investigation was performed with a validated online tool with a high technical & ethical standards. The methods are well described with detail & information to replicate.

Validity of the findings

The data is robust, well described statistically sound, & controlled. Conclusions are well stated, linked to original research question & limited to supporting results.

Additional comments

The authors have correctly addressed all my previous comments. I think their manuscript has been significantly improved and is now suitable for publication.
This is a very elegant and original study that will definitely add to the field.

Best,
Dr. Benjamin Rohaut